# Development of a Regional Gridded Runoff Dataset Using Long Short-Term Memory (LSTM) Networks

**Georgy Ayzel *** , **Liubov Kurochkina** , **Dmitriy Abramov** and **Sergei Zhuravlev**

State Hydrological Institute, 199004 Saint Petersburg, Russia; plathanthera@gmail.com (L.K.);
dmbrmv96@yandex.ru (D.A.); s.zhuravlev@hydrology.ru (S.Z.)
* Correspondence: ayzel@iwp.ru

**Abstract:** Gridded datasets provide spatially and temporally consistent runoff estimates that serve as reliable sources for assessing water resources from regional to global scales. This study presents LSTM-REG, a regional gridded runoff dataset for northwest Russia based on Long Short-Term Memory (LSTM) networks. LSTM-REG covers the period from 1980 to 2016 at a 0.5° spatial and daily temporal resolution. LSTM-REG has been extensively validated and benchmarked against GR4J-REG, a gridded runoff dataset based on a parsimonious regionalization scheme and the GR4J hydrological model. While both datasets provide runoff estimates with reliable prediction efficiency, LSTM-REG outperforms GR4J-REG for most basins in the independent evaluation set. Thus, the results demonstrate a higher generalization capacity of LSTM-REG than GR4J-REG, which can be attributed to the higher efficiency of the proposed LSTM-based regionalization scheme. The developed datasets are freely available in open repositories to foster further regional hydrology research in northwest Russia.

**Keywords:** runoff; modeling; reanalysis; dataset; neural networks; LSTM; northwest Russia





## 1. Introduction

Water is one of the most important natural resources on Earth. Its availability determines human development in many fields, such as water management, agriculture, and energy production [1]. However, water is also a cause of devastating natural disasters, such as floods and droughts [2,3]. Thus, monitoring and predicting available water resources are essential for many parties due to their high relevance for water management and planning and development of water-related risk mitigation strategies [1,4,5]. Furthermore, these issues become increasingly important due to climate change that amplifies the hydrological cycle intensity, affecting the frequency and magnitude of extreme events [2,6].

Due to its high societal [1] and scientific [4,7] relevance, river flow remains one of the most well-monitored components of the terrestrial water cycle [4,7–9]. However, the number of river flow gauges have gradually declined since the 1980s mainly due to budgetary constraints and political instabilities [4,7,10,11]. Thus, that puts the strong focus of the research community on the development of alternative yet robust techniques for water resources assessment and future projections when shrinking observational networks [4,8,12].

To date, gridded datasets provide a solid basis for spatially and temporally consistent estimates of relevant variables describing heat, water, and energy fluxes at every location within a spatial domain [13,14]. While gridded estimates of atmospheric variables, such as air temperature and precipitation, have proven their value for both scientists and practitioners [15,16], gridded datasets of runoff have recently begun to gain popularity [17]. During recent years, the international model intercomparison projects, such as earth2observe [18], or ISIMIP [19], as well as several individual research groups worldwide [8,12,20–22], provided both global and regional gridded estimates of runoff that have proven their reliability and value [17,23].

These gridded runoff datasets can be classified into two groups. The first group (e.g., datasets from earth2observe, or ISIMIP projects) is based on the utilization of standard process-based hydrological models and parameter regionalization techniques (e.g., spatial proximity, physical similarity, and regression-based approaches). The second group is based on the utilization of data-driven models. To date, there are two successful examples of data-driven-based datasets—E-RUN [8], and GRUN [12]. Both datasets are based on the Random Forest model [24], which has gained popularity in hydrology during the last decade [25]. Besides, there are other prospective data-driven models and techniques that have a prominent potential for gridded datasets development, e.g., spatiotemporal data fusion [26]. However, open datasets based on them are not yet readily available.

The first studies that utilize artificial neural networks (ANNs) for runoff prediction date back to 1990s [27]. Recently, deep learning, a field of machine learning investigating the potential of ANNs comprised of many computational layers, has emerged as a revolutionary technology generating new and improved capabilities for data-driven model development [28,29]. Deep learning models, in particular those based on so-called Long Short-Term Memory Networks (LSTM), have already shown a prominent value in many hydrological applications, such as runoff modeling [30–32], prediction in ungauged basins [33,34], and forecasting [35].

In the recent study, Gauch et al. [36] showed that LSTM-based models outperform tree-based models when longer observational periods for the larger sets on river basins are available for model training (calibration). In the presented study, we want to capitalize on the potential of LSTM and develop LSTM-REG—the first LSTM-based regional gridded runoff dataset for northwest Russia. To investigate the reliability of the proposed development workflow and the efficiency of the provided runoff estimates, we compare the LSTM-REG dataset with GR4J-REG—the dataset, which is based on GR4J hydrological model [37] concerted with the proximity-based parameter regionalization technique [22]. In our previous study [22], we showed that the dataset which was developed based on the same methodological basis as that for GR4J-REG outperformed both regional (E-RUN) and global gridded runoff datasets. Thus, GR4J-REG provides a solid basis against the newly developed LSTM-REG dataset is benchmarked. In summary, the objective of this study is to develop and then benchmark two types of gridded runoff datasets—process-based (GR4J-REG) and (novel) data-driven (LSTM-REG)—and then try to learn from their side-by-side comparison. To the best of our knowledge, this is the first time that the LSTM-based gridded runoff dataset is developed and comprehensively benchmarked in the context of regional hydrological modeling.

The paper is organized as follows. In Section 2, we describe the data sources for streamflow observations and meteorological forcing utilized in the presented study. Section 3 describes the methods we used for a regional gridded dataset development and describes their performance evaluation. We report the results and discuss them in various contexts in Section 4. Section 5 provides a summary and conclusions. In Section 6, we provide references to the open repositories where the developed gridded datasets (GR4J-REG and LSTM-REG) are readily and freely available.

## 2. Data

### 2.1. Streamflow Observations

The observational streamflow data is provided by the Regional Revised River Runoff Reanalysis (R5) project [22]. All the data was compiled from Russian archive streamflow data in annual digests (hydrological yearbooks), as well from the website of the Automated Information System for State Monitoring of Water Bodies (AIS; https://gmvo.skniivh.ru, last access: 15 December 2020) for river basins located within the particular R5 project-related geographical domain (25–57° E; 55–70° N). There are 275 unregulated and relatively undisturbed river basins in the compiled version of the R5 dataset (Figure 1), for which streamflow observations are available at a daily temporal resolution for the period from

1979 to 2016. The river basins are unevenly distributed over the study area, as they belong to the basins of the Barents, White and Baltic Seas.

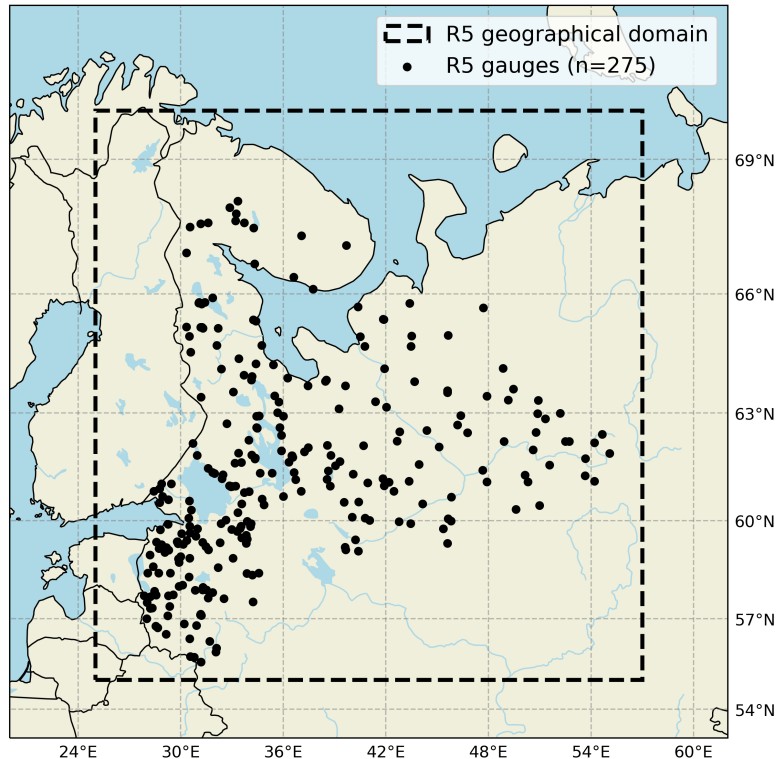

**Figure 1.** Gauge stations from the compiled dataset of streamflow observations of the R5 project.

*2.2. Meteorological Forcing Data*

Gridded observations of precipitation and air temperature data are obtained from the EWEMBI (EartH2Observe, WFDEI and ERA-Interim data Merged and Bias-corrected for ISIMIP) dataset [38–40]. EWEMBI dataset has global coverage for the period 1979–2016 with daily temporal resolution and $0.5° \times 0.5°$ spatial resolution. Potential evaporation was calculated based on a temperature-based equation proposed by Oudin et al. [41] using EWEMBI air temperature data. The EWEMBI dataset has been successfully used as a reliable source of meteorological forcing data for a wide range of hydrological models, and river basins worldwide [21,42,43].

**3. Methods**

*3.1. Hydrological Models*

3.1.1. GR4J

In the presented study, we used the conceptual lumped hydrological model GR4J [37], which was coupled with the Cema–Neige snow accounting routine [44,45]. The model GR4J-CemaNeige (referred hereinafter as GR4J, for simplicity; Figure 2) has been selected based on its availability, and simplicity of implementation [46] yet high efficiency in runoff predictions across the globe [16,22,47–49]. Moreover, it also demonstrated encouraging results in regionalization studies [16,50,51]. The GR4J model has a lumped structure with six parameters (two of which are related to the Cema–Neige snow model), runs at a daily time step, and requires daily time series of air temperature (T), precipitation (P), and potential evaporation (PE) as inputs.

GR4J model parameters are calibrated against streamflow observations in an automated manner using the global optimization algorithm of differential evolution [52] with the Kling–Gupta efficiency (KGE; Equation (1) [53]) as an objective function algorithm tries to maximize. In this way, the GR4J model is independently calibrated for every individual

considered river basin. We calibrate GR4J against the entire period of available streamflow observations to obtain a more robust set of optimal model parameters [47].

$$KGE = 1 - \sqrt{(R-1)^2 + (\alpha - 1)^2 + (\beta - 1)^2},$$ (1)

where $R$ is the correlation component represented by Pearson's correlation coefficient, the variability component $\alpha$ by the ratio of the estimated and observed standard deviations, and the bias component $\beta$ by the ratio of estimated and observed means (Equations (2) and (3)).

$$\alpha = \frac{\sigma_{sim}}{\sigma_{obs}}$$ (2)

$$\beta = \frac{\overline{Q_{sim}}}{\overline{Q_{obs}}}$$ (3)

$\sigma_{sim}$ and $\sigma_{obs}$ are standard deviation in simulations and observations, $\overline{Q_{sim}}$ and $\overline{Q_{obs}}$ are mean simulated and observed runoff. KGE is positively oriented and not limited at the bottom: a value of 1 represents a perfect correspondence between simulations and observations. According to Knoben et al. [54], KGE > −0.41 can be considered as skillful against mean flow benchmark and KGE > 0.3 can be considered as behavioral. According to Ratner [55], (absolute) R values between 0.7 and 1.0 indicate a strong linear relationship, values between 0.3 and 0.7—moderate relationship—and values between 0 and 0.3—weak linear relationship through a firm linear rule.

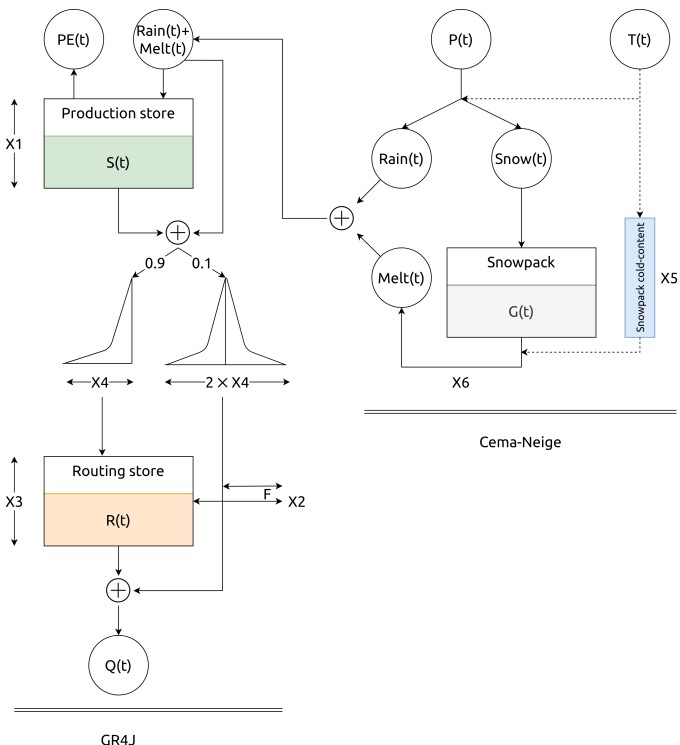

**Figure 2.** GR4J hydrological model structure.

### 3.1.2. LSTM

In this study, we consider the Long Short-Memory Network (LSTM)—the type of Artificial Neural Networks (ANN) which had been recently suggested for hydrological modeling [30]. We use a so-called three-layer topology, which includes one input, one output, and one computational layer. The first layer (input) propagates input variables such as temperature, precipitation, and potential evaporation to the second layer. The second layer (here—LSTM) receives a vector of input variables and then performs respective

computations. The second layer's output propagates further to the third (output) layer, which is always a dense (fully-connected) layer with one neuron and a linear activation function. The purpose of the third (output) layer is to set up a connection between the second layer's multidimensional output and output variable (here—runoff).

The LSTM network topology is a special case and an advanced successor of the Recurrent Neural Network (RNN). The latter has been specifically designed to utilize the sequential order of input variables by using a recurrent layer as a computational core. Thus, computations in a recurrent layer are performed sequentially from the first to the last time step of the input variables. To overcome the problem of RNN of learning, long-term dependencies in data which were aroused in several studies [56], the LSTM cell (Figure 3A) has been introduced [57].

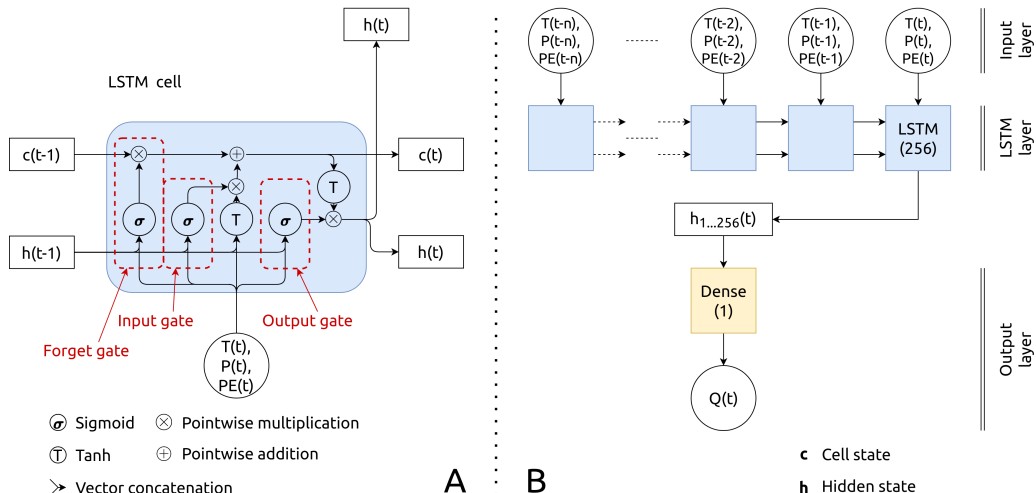

**Figure 3.** (**A**) The internal operations and activation functions of a Long Short-Term Memory (LSTM) cell, (**B**) a general representation of the LSTM model.

The LSTM cell (Figure 3A) utilizes a complex logic of internal computations by using two state variables, namely hidden ($h$) and cell ($c$) state variables, as well as three "gates"—forget ($f_t$), input ($i_t$), and output ($o_t$) [57]. Thus, there is a sequential update of the both hidden and cell state variables while moving from the first ($t - n$) to the last ($t$) step of the input sequence. The last updated hidden state variable $h$ (at time $t$) connects to the output (third) layer that features a single neuron and a linear activation function (Figure 3B). The detailed information of the internal computations in the LSTM cell, as well as further details on LSTM networks in a hydrological context can be found in Kratzert et al. [30].

The hyperparameters of this design are the number of last hours $n$ (the length of the input sequence) and the length of the state variables, both for hidden and cell states. Following the recent studies [31,33], we decided to keep $n$ of 365 (days), and the length of both state variables equal to 256 (cells). We have found that this setting shows promising results in a number of previous studies [32,34]. In total, the LSTM model has 266,497 parameters. In contrast to the GR4J model, the high number of parameters in the LSTM model prevents the utilization of global optimization algorithms due to the high computational costs. Instead, backpropagation is used for the efficient calibration of ANNs [58], updating model parameters through the calculation of the gradient of the objective function [59]. In the presented study, for the optimization of LSTM parameters, we used the Adam algorithm for stochastic gradient descent-based optimization with default hyperparameters [60], the mean squared error as a loss function, the batch size of 256, and the maximum number of epochs of 100. However, to reduce calibration time and control model overfitting, we employ the early stopping technique that interrupts the calibration procedure while there is no improvement in the hold-out test period for ten epochs. The respective hold-out test period is randomly assigned and consists of 25% of the entire calibration data length. For efficient calibration, both meteorological variables and runoff have been scaled by

subtracting the mean and dividing by the standard deviation [30,61]. Further details on the ANN calibration procedure can be found in Kratzert et al. [30].

In contrast to the GR4J model, which is calibrated individually for each considered river basin, the LSTM model here is calibrated simultaneously for all the considered river basins. In this way, the LSTM model explicitly accounts for the regional patterns of rainfall-runoff processes [34].

### 3.2. Gridded Runoff Dataset Development Workflows

3.2.1. Selection of Donor Basins

In the presented study, the development of gridded runoff datasets begins with the selection of donor basins. The corresponding selection is based on the calibration score in terms of KGE (Equation (1)) and basin area. If the basin has KGE $\geq$ 0.75, it is presumed to have good-quality observed runoff, and meteorological forcing data [54]. The basin area should be from 10 to 10,000 km$^2$. This interval ensures the reliability of forcing data [62] and help to minimize the effects of channel routing [62,63]. Thus, the basin that has both KGE $\geq$ 0.75 and an area of 10–10,000 km$^2$, could serve as a reliable donor basin for the regionalization scheme [22,50,62].

In this way, the GR4J model has been calibrated for each available R5 basin. Then, we select only those basins that have KGE $\geq$ 0.75 and $10 \leq$ area $\leq$ 10,000 km$^2$ as donor basins. Further development of both gridded runoff datasets is based on the same set of donor basins.

3.2.2. GR4J-REG

The proposed workflow (Figure 4) for the development of a gridded runoff reanalysis dataset is based on the workflow introduced by Ayzel et al. [22]. That workflow has proved its reliability and high efficiency, outperforming both regional and global gridded runoff datasets for the territory of northwest Russia [22].

The workflow is based on the utilization of the GR4J hydrological model with the parsimonious proximity-based regionalization scheme (Figure 4). In summary, after the basin-scale calibration and the selection of donor basins, we use a spatial proximity-based (nearest-neighbor) model parameter regionalization approach to attribute each grid cell inside the R5 geographical domain (1920 grid cells in total considering a spatial resolution of 0.5°) to the ten nearest donor basins. Then, to obtain an ensemble of simulated runoff time series for each grid cell, we run GR4J with the ten attributed optimal model parameter sets. Finally, we compute an ensemble mean of ten simulated runoff time series for each respective grid cell. In this way, we obtain GR4J-REG—the gridded runoff dataset at daily temporal resolution and 0.5° spatial resolution for the entire R5 geographical domain based on the GR4J hydrological model and proximity-based regionalization technique. GR4J-REG spans the period from 1979 to 2016.

3.2.3. LSTM-REG

Figure 5 demonstrates the LSTM-REG development workflow. In regard to hydrological model development (yellow box in Figure 5), the utilized workflow does not differ from that for establishing any data-driven model for runoff prediction at regional, continental, or global scale [8,12,31,34]. In the presented study, a single regional LSTM model is calibrated based on the observational data from all the donor basins. To ensure the comparability and consistency with the GR4J-REG dataset, for each grid cell of which an ensemble of 10 simulated runoff time series is computed, ten different regional LSTM models were calibrated. The stochasticity in LSTM initialization and randomization of training and hold-out testing data while a single LSTM calibration, results in different yet efficient models [31]. Then, to obtain an ensemble of simulated runoff time series for each grid cell, we run the ten regional LSTM models. Finally, we compute an ensemble mean of ten simulated runoff time series for each respective grid cell. In this way, we obtain LSTM-REG—the gridded

runoff dataset at daily temporal resolution and 0.5° spatial resolution for the entire R5 geographical domain based on Long Short-Term Memory networks.

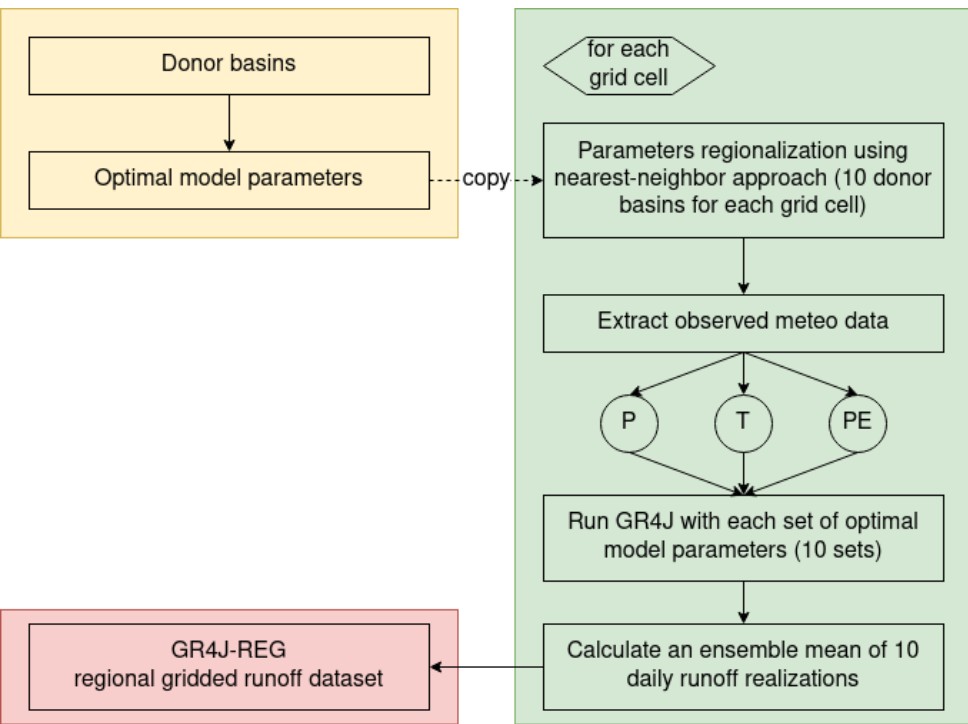

**Figure 4.** Schematic representation of the proposed workflow for the development of the GR4J-REG dataset.

In contrast to GR4J-REG, LSTM-REG spans a shorter period (1980–2016) because LSTM by design needs the preceding 365 days of meteorological data as input for runoff prediction. Thus, we cannot simulate the first year of the available period.

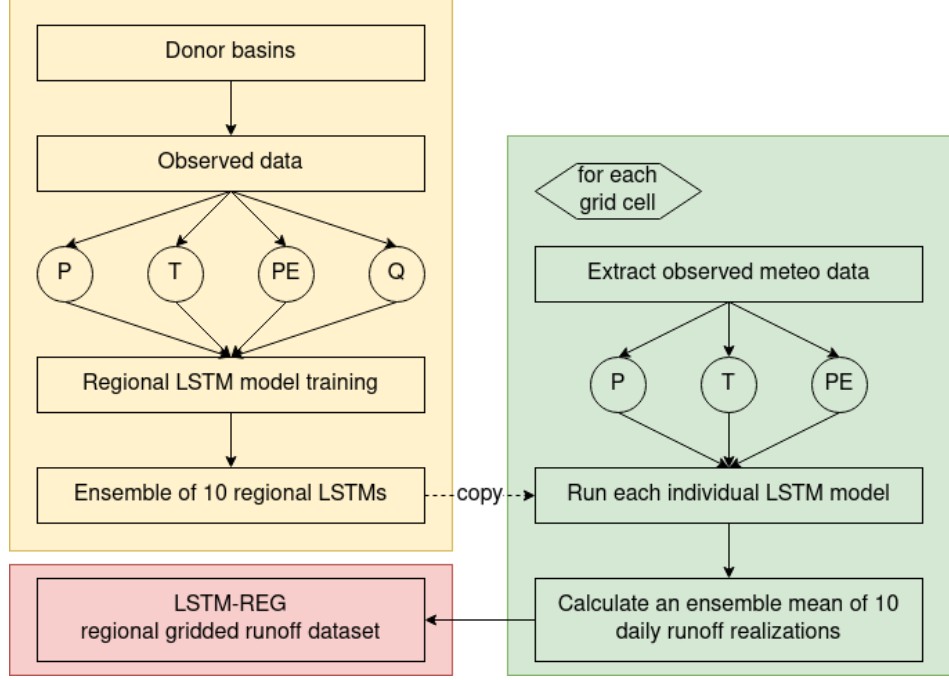

**Figure 5.** Schematic representation of the proposed workflow for the development of the LSTM-REG dataset.

### 3.3. Performance Evaluation

Since considered datasets have different temporal spans, first, we made them consistent for the period from 1 January 1980 to 31 December 2016.

For the evaluation of the developed datasets' performance at the basin scale, we use those basins that do not pass the selection criteria for donor basins (Section 3.2.1), but where the GR4J model demonstrates at least behavioral efficiency (KGE $\geq$ 0.3; [54]). Following the recent studies [15,17,64–66], for the evaluation set of basins, we assess the efficiency of the developed datasets in comparison with basin-scale calibration results in terms of KGE (Equation (1)) and its components—correlation R, variability $\alpha$ (Equation (2)), and bias $\beta$ (Equation (3))—at daily temporal resolution. The KGE metric has been chosen due to the fact that the hydrological community has recently been changing its preference to KGE [17,54,64,67] instead of the Nash–Sutcliffe efficiency [68], which, although widely used, has a number of significant drawbacks [69,70].

Another important component of the evaluation is whether the LSTM-REG dataset adds value or have skill compared to the benchmark GR4J-REG dataset. Thus, a prediction skill (Equation (4)) can be assessed by the direct comparison of the prediction (LSTM-REG), and benchmark (GR4J-REG) accuracy for a given efficiency metric [71].

$$Skill_{metric} = \frac{A_{prediction} - A_{benchmark}}{A_{perfect} - A_{benchmark}}, \tag{4}$$

where $A_{prediction}$ is the prediction accuracy in terms of some efficiency metric, $A_{benchmark}$ is the benchmark accuracy in terms of the same efficiency metric, and $A_{perfect}$ is the perfect accuracy for this metric, i.e., 1 for KGE and R. Skill score values are assigned qualitative descriptions, such as having positive (skill > 0) and negative (skill < 0) skill [72,73].

## 4. Results and Discussion

### 4.1. GR4J Model Calibration and Donor Basin Selection

Figure 6 shows the calibration efficiency of the GR4J model driven by EWEMBI meteorological reanalysis for the entire available period of runoff observations for all the basins (Figure 1) under consideration. The median value for daily KGE is 0.86, with the interquartile range (IQR; IQR = Q75−Q25) of 0.09. The majority of river basins (265 out of 275) shows KGE higher than 0.3, which is a threshold for behavioral predictions [54]. Thus, the GR4J model (again) proves its high efficiency for runoff predictions in basins located in various geographical regions [16,22,50].

Two hundred forty-three (243) basins have KGE higher than 0.75, which is a threshold for the selection of donor basins (Section 3.2.1). Two hundred eleven (211) out of these 243 basins have a basin area from 10 to 10,000 km$^2$. Thus, these 211 basins have been selected as donors for the development of gridded runoff datasets. Fifty-four (54) out of the remaining basins have KGE higher than behavioral (KGE $\geq$ 0.3) and have been selected for the evaluation. Figure 7 shows the spatial distribution of the donor and the evaluation set of basins. Both sets have a relatively uniform distribution in space. Therefore, we argue that the performance evaluation of gridded runoff datasets will be representative for the considered geographical domain.

The percentage of suitable donor basins in the R5 dataset is 77% (211 of 275). The use of softer criteria for the selection of donor basins in this study (KGE $\geq$ 0.75 and $10 \leq$ area $\leq$ 10,000 km$^2$) provides an opportunity for a significant increase in their relative number in comparison to the study of Beck et al. [62], in which only 38% (674 of 1787) of basins were selected for the regionalization procedure. The decision on donor basin selection criteria that results in an increase of the number of reliable donor basins has a distinct positive effect on the resulting performance of the developed runoff datasets, as discussed in Ayzel et al. [22].

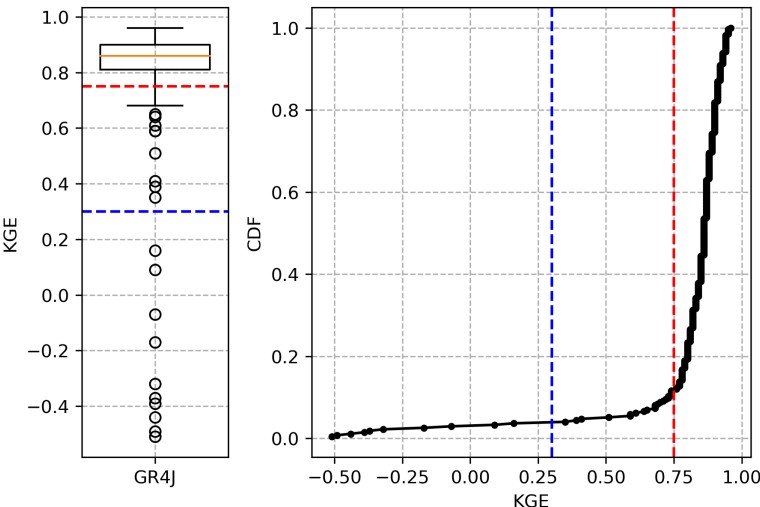

**Figure 6.** Basin-scale calibration results for the GR4J model. The box plot (left plot) and cumulative density function (CDF, right plot) demonstrate the distribution of calibration efficiency for all 275 R5 basins in terms of KGE. The boxplot box represents the Interquartile Range (IQR, the difference between the 25th and 75th quantiles); the whiskers represent ±1.5 IQR from the 25th and 75th quantiles, respectively; the yellow line denotes the median value. The blue dashed line represents the threshold for behavioral predictions (KGE ≥ 0.3); the red dashed line—the threshold used for the selection of donor basins (KGE ≥ 0.75).

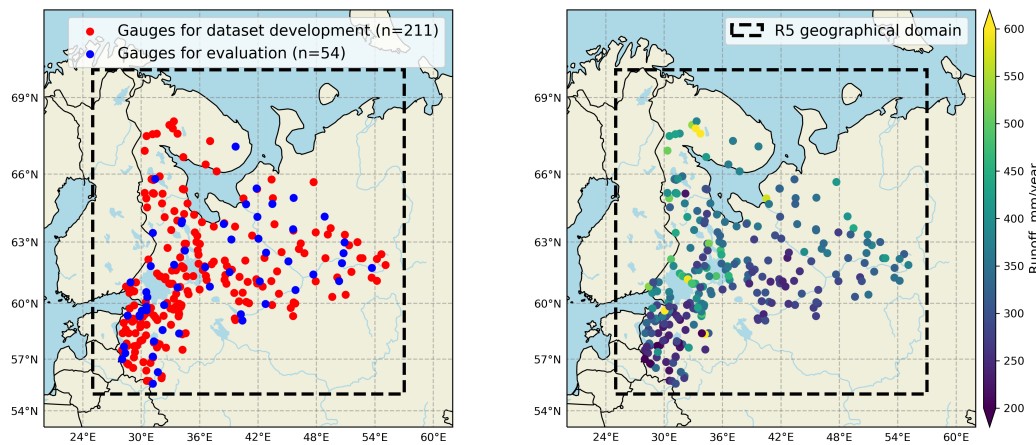

**Figure 7.** Spatial distribution of gauges that have been used for dataset development (i.e., donor basins) and performance evaluation (**left plot**), and the corresponding values of mean annual runoff (**right plot**).

## 4.2. Performance of the Developed Gridded Runoff Datasets

The critical characteristic of the quality of any regionalization scheme (and resulting gridded dataset) is the difference in efficiency between runoff prediction with calibrated, and regionalized parameters [22]. This difference in efficiency represents the ability of the proposed technique to be used for runoff reconstruction [50,62,74]. Thus, the obtained margin in prediction efficiency between developed gridded runoff datasets and basin-scale calibration results characterizes a generalization capacity of the developed datasets: the lower the gap, the higher the generalization capacity and vice versa. Figure 8 demonstrates the results of the performance evaluation of the developed GR4J-REG and LSTM-REG datasets in comparison with basin-scale calibration results of the GR4J model for the evaluation set of basins in terms of KGE and Pearson's correlation coefficient R.

In terms of KGE calculated for daily predictions, the median efficiency of GR4J basin-scale calibration is 0.86 with an IQR of 0.22. For GR4J-REG and LSTM-REG datasets, the

median KGE are 0.52 (IQR = 0.71) and 0.58 (IQR = 0.79), respectively. Thus, there is a considerable drop in runoff prediction efficiency between the developed gridded runoff datasets in comparison to GR4J basin-scale calibration results. The corresponding differences between developed datasets and basin-scale model calibration are less pronounced yet considerable in terms of correlation coefficient R. The median R are 0.86 (IQR = 0.21), 0.8 (IQR = 0.28), and 0.82 (IQR = 0.3) for GR4J, GR4J-REG, and LSTM-REG, respectively. The results obtained for the GR4J-REG and LSTM-REG datasets are consistent with previous regionalization studies performed in different geographical regions regarding the range of prediction performance (see, e.g., Parajka et al. [75] for a review). Thus, the proposed techniques for gridded runoff dataset development are in line with the state-of-the-art in the field.

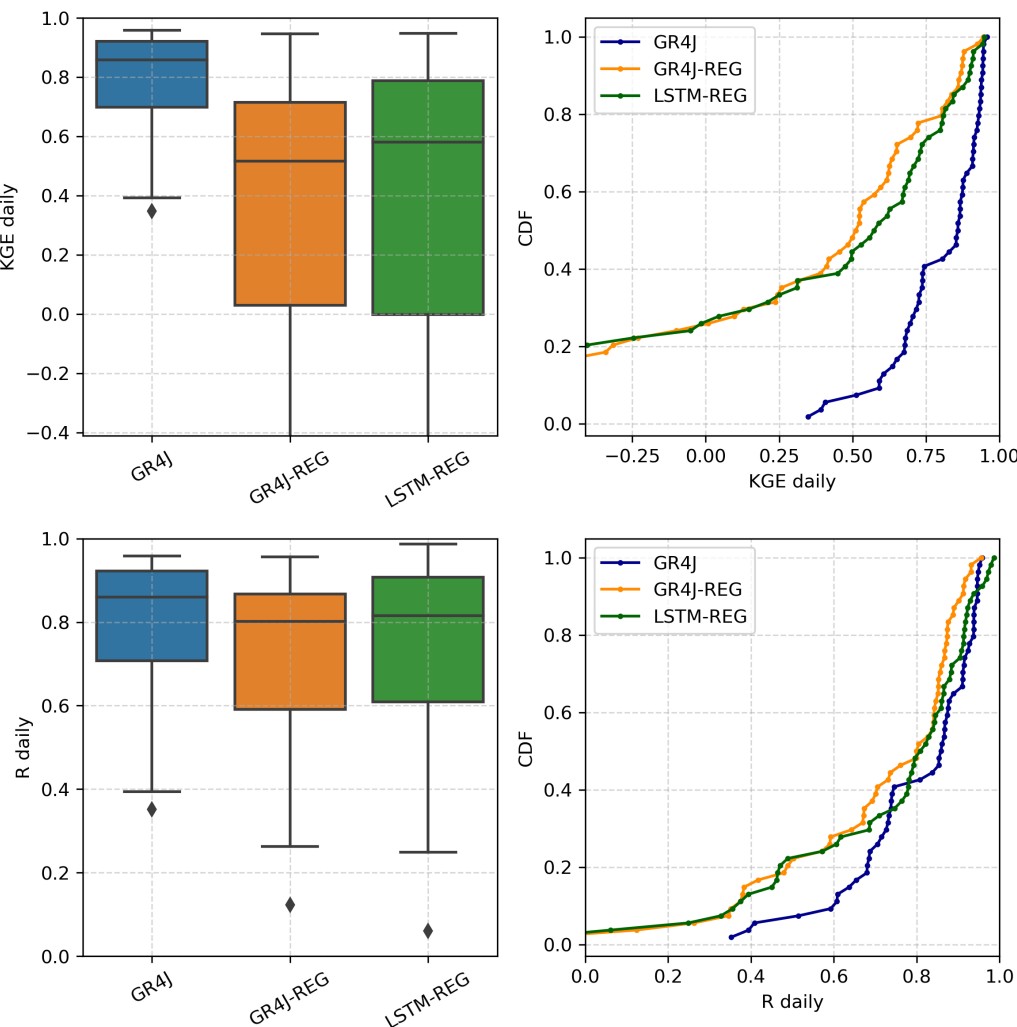

**Figure 8.** Performance evaluation results of GR4J-REG and LSTM-REG gridded runoff datasets in comparison with the results of GR4J basin-scale calibration in terms of the Kling–Gupta efficiency (KGE, **top row**) and Pearson's coefficient of correlation (R, **bottom row**). The results are presented in the form of box plots (**left column**) and cumulative density functions (CDF, **right column**).

LSTM-REG has a higher mean and median KGE and R than those for GR4J-REG. Welch's *t*-test also confirms that the difference between datasets' means is significant. In the evaluation set, each third basin falls under the KGE's behavioral threshold: there are 19 and 18 basins with KGE < 0.3 for GR4J-REG and LSTM-REG, respectively. While for basins with low, non-behavioral (KGE < 0.3) efficiencies, the difference between datasets is marginal, the value of LSTM-REG against GR4J-REG becomes more pronounced for higher

KGEs (Figure 8, top right plot). Similar behavior corresponds to the correlation coefficient R: while for basins with lower than moderate correlation values (R < 0.7), the differences between datasets are also marginal, LSTM-REG capitalizes more for higher Rs (Figure 8, bottom right plot). The obtained results are consistent with those obtained in our previous study [22] and demonstrate the tangible value of regional datasets for runoff assessment.

Figure 9 demonstrates the results of the performance evaluation of the developed GR4J-REG and LSTM-REG datasets in comparison with basin-scale calibration results of the GR4J model for the evaluation set of basins in terms of two KGE components (Equations (2) and (3)): the variability $\alpha$ (left plot) and bias $\beta$ (right plot). Both datasets demonstrate considerable differences between each other and compared to basin-scale calibration in terms of both metrics. For GR4J-REG, the median $\alpha$ and $\beta$ are almost 1, with IQRs are lower than 0.02, which are solid indicators of reliable calibration results in terms of reproducing the climatological characteristics of runoff (see also Section 4.1). For GR4J-REG and LSTM-REG, the median $\alpha$ are 1.31 and 1.18, and the median $\beta$ are 1.05 and 1.09, respectively. Results demonstrate that, on average, LSTM-REG provides runoff estimates that have lower variability component (the ratio of the estimated and observed standard deviations), but almost similar bias (the ratio of estimated and observed means) than those for GR4J-REG. Both datasets' IQRs of $\alpha$ and $\beta$ tend to the region of positive anomalies, i.e., for the most of analyzed basins, runoff simulations have higher variability and bias than observations. In terms of bias component, the results are incosistent with the previous studies that show that both regional and global runoff datasets have a tendency to underestimate both mean and high river flows [8,12,17,76,77]. Thus, the obtained results demonstrate a higher generalization capacity of LSTM-REG compared to GR4J-REG, which can be attributed to the higher efficiency of the proposed LSTM-based regionalization scheme.

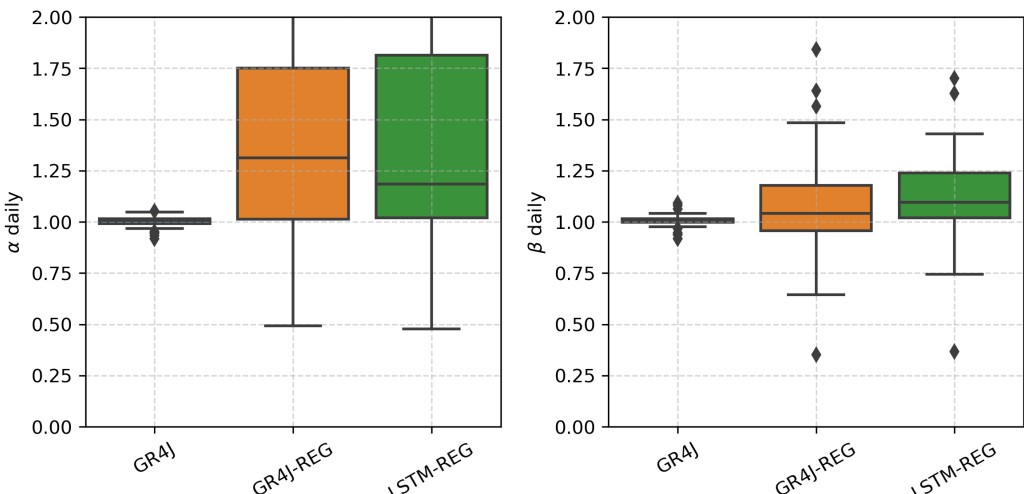

**Figure 9.** Performance evaluation results of GR4J-REG and LSTM-REG gridded runoff datasets in comparison with the results of GR4J basin-scale calibration in terms of the variability component $\alpha$ (**left plot**) and bias component $\beta$ (**right plot**) of the Kling–Gupta efficiency. The results are presented in the form of box plots.

Figure 10 shows the spatial distribution of KGE and its three components, namely, correlation R, variability $\alpha$, and $\beta$ for the evaluation set of basins. It can be seen that the GR4J model (Figure 10, left column) performs considerably worse in terms of correlation, which makes sense given that long-term climatological runoff statistics are easier to fit than daily dynamics. The calculated median values for the correlation, variability, and bias components of the KGE, expressed as $|R - 1|$, $|\alpha - 1|$, and $|\beta - 1|$, are 0.14, 0.01, and 0.01, respectively. Thus, these results indicate that improved correlation is of primary importance in terms of improving KGE scores for GR4J.

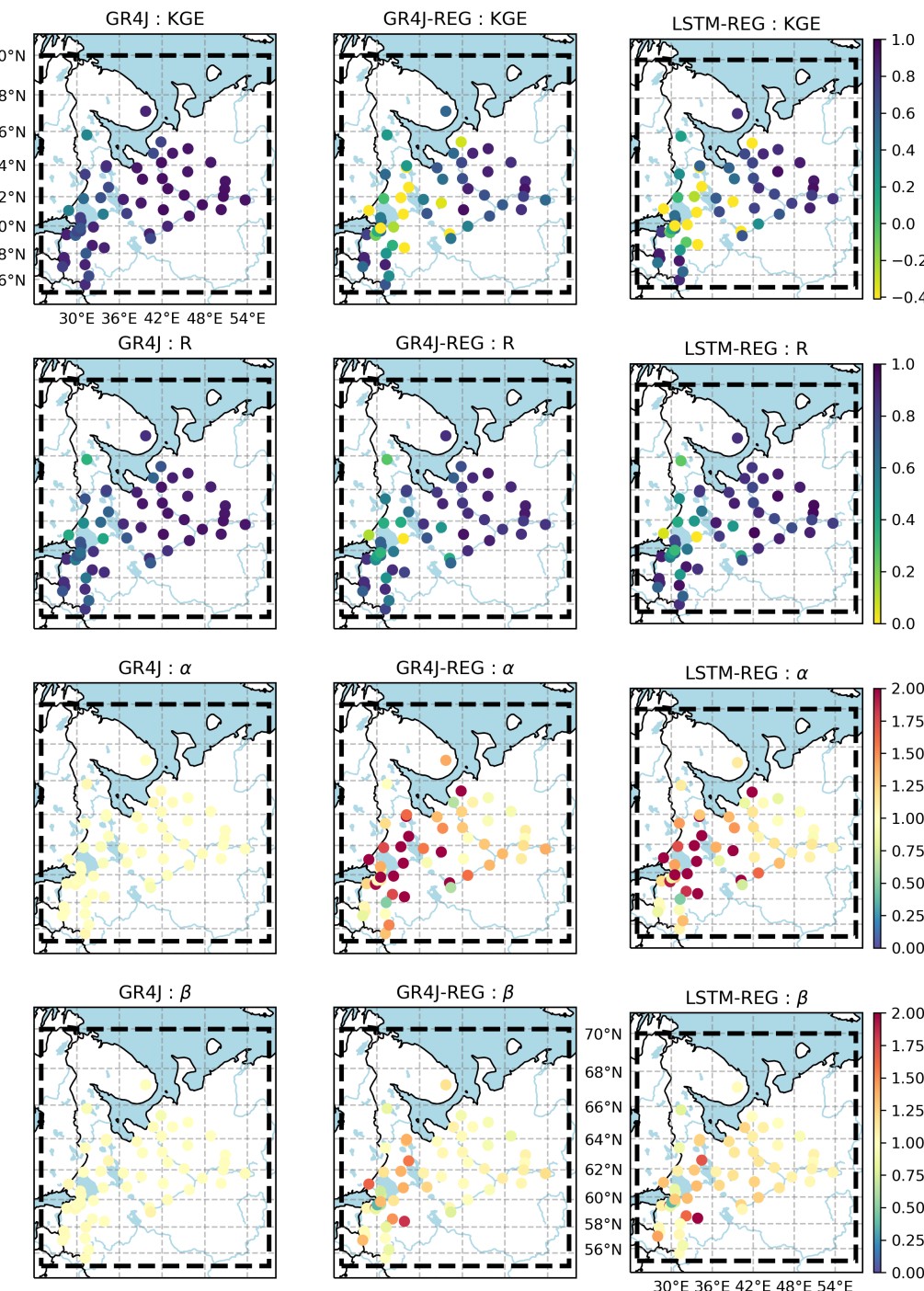

**Figure 10.** Performance evaluation results of GR4J-REG and LSTM-REG gridded runoff datasets in comparison with the results of GR4J basin-scale calibration in terms of the Kling–Gupta efficiency and its components: correlation R, variability $\alpha$, and bias $\beta$. The results are presented in maps that show a spatial distribution of the respective evaluation metrics.

In contrast to GR4J, GR4J-REG and LSTM-REG datasets show the different behavior that reflects the increased influence of both variability and bias components on KGE scores (Figure 10, middle and right columns). The calculated median values for KGE components, expressed as $|R - 1|$, $|\alpha - 1|$, and $|\beta - 1|$, are 0.2 (0.18), 0.31 (0.18), and 0.04 (0.09), for GR4J-REG and LSTM-REG (in parentheses), respectively. Thus, these results suggest that in order to get an improved KGE score, the two most important components to focus on are variability and correlation, followed by bias. For the GR4J-REG, the



variability's dominant influence may indicate the limitation of the proposed parsimonious regionalization technique that does not account for any information except spatial distance while attributing donor basins. LSTM-REG shows a higher median bias than that for GR4j-REG (Figure 9, right plot). However, the influence of the variability and correlation terms are twice as great as the bias term. That, in turn, suggests that for the developed gridded runoff datasets—GR4J-REG and LSTM-REG—improvements to the timing of runoff at the daily scale (dominating correlation scores) and runoff distribution (dominating variability scores) are more valuable than improvements to runoff totals (dominating bias scores). The presented results reveal the crucial role of improvements to variability component needed for the development of reliable runoff datasets in addition to the dominant role of correlation described earlier for daily precipitation datasets [66].

Figure 11 shows the skill of LSTM-REG against GR4J-REG for the evaluation set of basins. The skill is computed according to Equation (4) based on daily KGE. The results show that 35 of 54 analyzed basins (65%) have a positive (>0) skill. The spatial distribution of skill (Figure 11) shows that the positive skill is more pronounced for basins that are located closer to the domain boundaries. That effect can be attributed, on the one hand, to the limitation of the GR4J model parameters regionalization scheme that attributes the closest donor basins to a particular grid cell. In this way, for grid cells located near the domain's boundaries, the search for the corresponding donor basins was performed only in the "one way" direction (to the center from the border). On the other hand, that effect can be attributed to the advance of data-driven model calibration, which can utilize all the available information from donor basins disregarding their spatial location or temporal span.

There was no clear pattern in the interconnection between the skill (in terms of using the KGE metric) of LSTM-REG against GR4J-REG and basin area (Figure 11) for areas up to $10^4$ km$^2$. However, for a group of large basins ($10^4$ km$^2$–$10^5$ km$^2$), there are some basins that benefit the most from LSTM-REG (with the highest skill), as well as some other basins for those GR4J-REG provide more skillful predictions. LSTM-REG outperforms GR4J-REG predictions for the largest five basins (from $10^5$ km$^2$).

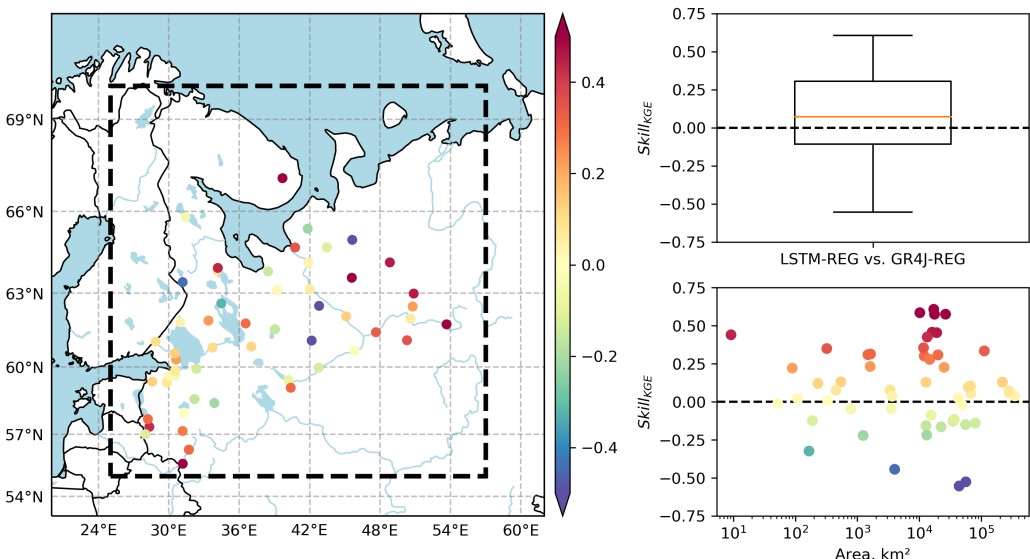

**Figure 11.** The KGE-based skill of LSTM-REG against GR4J-REG for the evaluation set of basins: map (left plot) represents the spatial distribution of skill, box plot (top right plot) represents the distribution properties of skill, scatter plot (bottom right plot) demonstrates the interconnection between skill and basin area. The skill is computed according to Equation (4) based on daily KGE.

Figure 12 shows the skill of LSTM-REG against GR4J-REG that is computed based on daily Pearson's correlation (R) for the evaluation set of basins. While the correlation-

based skill is visually more pronounced than KGE-based, the results show that the same number of basins (35 of 54, 65%) have a positive (>0) skill. The spatial distribution of correlation-based skill is similar to that for the KGE-based skill: basins located closer to domain boundaries benefit more from LSTM-REG than GR4J-REG. In contrast to the KGE-based skill (Figure 11), correlation-based skill shows that GR4J-REG outperforms LSTM-REG predictions for four out of five largest basins (from $10^5$ km$^2$, Figure 12). That can be attributed to the increasing role of variability and bias components in KGE scores obtained for the evaluation set's largest basins.

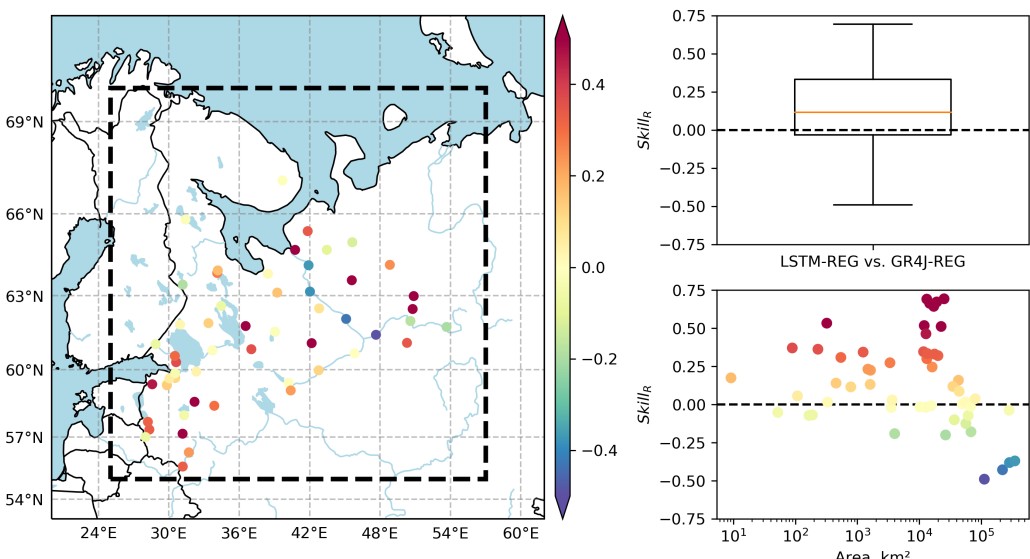

**Figure 12.** The correlation-based skill of LSTM-REG against GR4J-REG for the evaluation set of basins: map (**left plot**) represents the spatial distribution of skill, box plot (**top right plot**) represents the distribution properties of skill, scatter plot (**bottom right plot**) demonstrates the interconnection between skill and basin area. The skill is computed according to Equation (4) based on daily R.

In addition, we calculated the skill of LSTM-REG and GR4J-REG against GR4J basin-scale calibration. The results show that GR4J-REG outperforms GR4J only for a single basin with a skill of 0.07. In contrast to GR4J-REG, LSTM-REG outperforms GR4J for seven of 54 basins (13%) with a mean skill of 0.22. This confirms the results obtained by Kratzert et al. [33] and Ayzel et al. [34] who showed that the out-of-sample LSTM model outperforms the standard hydrological model (SAC-SMA and GR4J, respectively) for prediction in ungauged basins in 58% and 42% of basins, respectively. Thus, the obtained results once again indicate the ample potential of machine learning-based techniques for hydrological modeling [8,12,28–34,36,78,79].

In summary, the results show that LSTM-REG—a regional gridded runoff dataset based on Long Short-Term Memory networks—provides efficient, reliable, and skillful runoff estimates for the independent evaluation set of basins. Thus, LSTM-REG and the underlying development workflow provide a solid basis for further research of runoff characteristics at the regional scale.

## 5. Conclusions and Outlook

Gridded datasets provide spatially and temporally consistent runoff estimates that are reliable sources for an assessment of water resources from regional to global scales [16–18,22]. In the study presented herein, we introduced the new workflow for the regional gridded runoff dataset development, which is based on the emerging data-driven model from the field of deep learning—LSTM networks. Based on the introduced development workflow, the regional gridded runoff dataset LSTM-REG has been developed. LSTM-REG has daily temporal resolution and 0.5° spatial resolution. The performance of LSTM-REG has been

extensively validated at basin scale in comparison to a solid benchmark—the GR4J-REG dataset. The latter has been developed based on a previously introduced parsimonious yet efficient regionalization scheme, and the GR4J hydrological model [22].

We comprehensively evaluated the developed gridded runoff datasets' performance against available observations for the evaluation set of 54 river basins as reference. The evaluation was carried out at a daily scale for the period 1980–2016 using the Kling–Gupta efficiency (KGE), an objective performance metric combining correlation (R), variability ($\alpha$), and bias ($\beta$) components. While both datasets provide runoff estimates with reliable prediction efficiency, LSTM-REG outperforms GR4J-REG for most basins in the independent evaluation set. Thus, the results demonstrate a higher generalization capacity of LSTM-REG than GR4J-REG, which can be attributed to the higher efficiency of the proposed LSTM-based regionalization scheme.

Analysis of the KGE components helped us reveal their relative importance in further developing gridded runoff datasets. Thus, results indicate that improved correlation is of primary importance, and improved variability and bias are of secondary importance in improving KGE scores for the basin-scale calibration of the GR4J hydrological model. In contrast, results for the developed GR4J-REG and LSTM-REG datasets suggest that the two most important components to focus on for further development are variability and correlation, followed by bias. The influence of the variability and correlation terms are twice as significant as the bias term. That, in turn, suggests that for the developed gridded runoff datasets, improvements to the timing of runoff at the daily scale (determining the correlation score) and runoff distribution (determining the variability score) are more valuable than improvements to runoff totals (affecting the bias score).

The developed LSTM-REG dataset could be utilized for several prospective research directions, such as prediction in ungauged basins [33,34] and large-scale and long-term assessment of available water resources in northwest Russia. For example, Figure 13 demonstrates the differences in the estimates of mean annual runoff calculated based on two datasets developed in the presented study—GR4J-REG and LSTM-REG. Learning from these differences could pave the way to a better understanding of hydrological processes and their dynamics at different spatial and temporal scales [80,81]. Furthermore, the developed ensemble of LSTM models could be further used for the development of gridded runoff projections for the XXI century. The required forcing data is readily available from the ISIMIP project [19]. However, the computational costs are still high and could limit the progress in the respective direction.

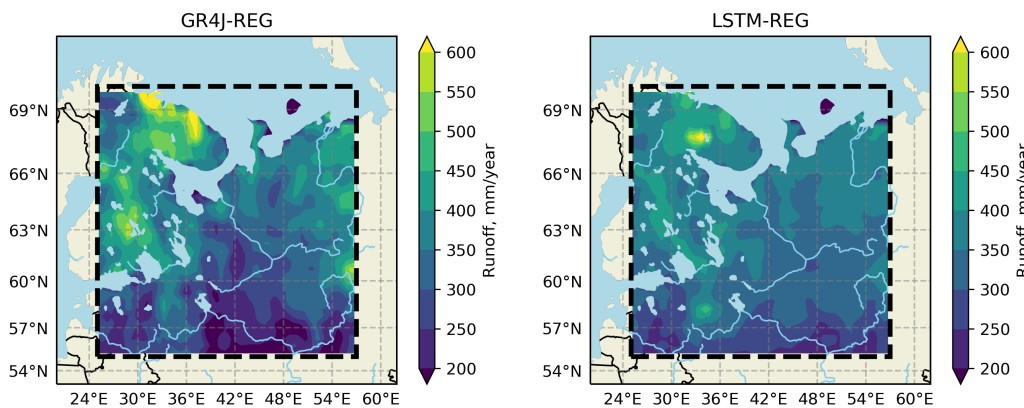

**Figure 13.** Mean annual runoff estimates for the developed gridded runoff datasets.

## 6. Data Availability

The GR4J-REG dataset is available at https://doi.org/10.5281/zenodo.4317312 (last access: 15 December 2020). The LSTM-REG dataset is available at https://doi.org/10.5281/zenodo.4317326 (last access: 15 December 2020).

**Author Contributions:** G.A. and S.Z. initiated this investigation. G.A., L.K., D.A. and S.Z. designed the study. G.A. developed the model code and performed the analysis. G.A. prepared the paper with contributions from all co-authors. All authors have read and agreed to the published version of the manuscript.

**Funding:** The reported study was funded by the Russian Foundation for Basic Research (RFBR) according to the research projects no. 19-05-00087 and no. 19-35-60005.

**Conflicts of Interest:** The authors declare no conflict of interest.

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
