# Peer review of "Development of a Regional Gridded Runoff Dataset Using Long Short-Term Memory (LSTM) Networks"

_hydrology, doi:10.3390/hydrology8010006_

Round 1

Reviewer 1 Report

This is an interesting and relevant contribution which adds to the growing interest in rainfall–runoff modelling using deep learning approaches, in particular LSTM. 

The paper is well written, logically laid out, and supported by results.

One of the main conclusions of the paper is that LSTM-REG outperforms the conceptual hydrological model GR4J-REG. The authors propose that such a performance can be attributed to the higher efficiency of the proposed LSTM-based regionalization scheme. 

In my opinion, however, this is the main critique of the work. Namely conceptual models have their own limitations. It can be argued that improved performance is not due to LSTM-based regionalization, but the limitations of conceptual models. The conclusions should therefore be re-stated. Furthermore, recently published works that utilise machine learning techniques show that it is possible to overcome limitations of both deterministic and conceptual models. See:

J Chadalawada et al, 2020, Hydrologically Informed Machine Learning for Rainfall‐Runoff Modeling: A Genetic Programming‐Based Toolkit for Automatic Model Induction, Water Resources Research 56 (4), e2019WR026933

HMVV Herath et al, 2020, Hydrologically Informed Machine Learning for Rainfall-Runoff Modelling: Towards Distributed Modelling, Hydrology and Earth System Sciences, 1-42

The authors should dedicate more attention to support their conclusions.

In addition to LSTM and other deep learning techniques, some authors rely on the use of computer vision for spatiotemporal data fusion, which may be yet another approach to the development of gridded runoff products:

S Jiang, et al, 2018, A computer vision-based approach to fusing spatiotemporal data for hydrological modeling, Journal of Hydrology 567, 25-40

Model performance evaluation: In my opinion, the metrics used to compare the model performance is rather limited, and could be expanded to incorporate additional and more insightful performance indices:

J Chadalawada, V Babovic, 2019, Review and comparison of performance indices for automatic model induction, Journal of Hydroinformatics 21 (1), 13-31

Finally, and from a more historical perspective, machine learning in hydrology (and in particular neural networks) have been quite popular since mid-1990s. It may be a good idea to refer to one of the foundational works:

Minns, A.W. and Hall, M.J., 1996, Artificial Neural Networks as Rainfall-Runoff Models. Hydrological Sciences Journal, 41, 399-417.

Reviewer 2 Report

The authors developed regional gridded runoff dataset in Northwest Russia using Long Short-Term Memory (LSTM) networks and the results show that LSTM-REG outperforms GR4J-REG for most basins in the independent evaluation set. Research is interesting and the deep learning method is novel and useful for a better understanding of hydrological processes. I suggest minor modification is needed before acceptance. The specific comments are below:

  1. In the introduction part, the authors refered two successful Random Forest model runoff dataset—E-RUN and GRUN datasets. In the result and conclusion part, what are advantage and disadvantage of LSTM-REG and GR4J-REG compared with these global dataset in Northwest Russia.
  2. Figure 7 show the spatial distribution of gauges that have been used for dataset development and performance evaluation. I wonder if the selection of gauges could affact the product quality. I suggest the authors add a map to show gauge-based multi-annual average runoff.

Reviewer 3 Report

This manuscript investigates the development of a regional gridded runoff dataset for northwest Russia based on Long Short-Term Memory (LSTM) networks. The topic is of paramount importance for hydrological applications also in areas different from that on which this study is focused. The paper is overall well written and to my opinion, it deserves to be published after minor revisions.

General comments:

To my opinion, there are only a few aspects that need to be improved in this paper, that I list here:

  1. the title should be changed, because if you are using “the”, then you have to define clearly what dataset are you developing (e.g., with the name). A solution can be the addition of “LSTM-REG”.
  2. the introduction should be clearly explaining the role of this paper in the current hydrological framework, better specifying its innovative elements. Furthermore, in the ending of the paragraph you should illustrate how the paper is structured, providing a guide for the reader.
  3. I would appreciate some insights on the applications of your findings on ungauged basins.

Specific comments:

  • Line 17: “:” are not necessary; you can simply add “such as”;
  • Line 19: “sustainable development and risk mitigation” are, to me, too generic concepts, that can be specified with more precision;
  • Lines 19-20: please, specify these aspects;
  • Line 21: why do you specify “(streamflow, discharge)”?
  • Line 32: please, use italic font for project names;
  • Lines 35-36: please, rephrase;
  • Line 37: is it necessary to report again citations? Maybe you can add others
  • Line 39: it is not well linked to other sentences;
  • Line 44: remove “,” after “-“.
  • Figures 2 and 3: please, improve font resolution of words in figures.

Kind Regards
